# Elusive Trans-Acting Factors Which Operate with Type I (Poliovirus-like) IRES Elements

**DOI:** 10.3390/ijms232415497

**Published:** 2022-12-07

**Authors:** Dmitry E. Andreev, Michael Niepmann, Ivan N. Shatsky

**Affiliations:** 1Shemyakin-Ovchinnikov Institute of Bioorganic Chemistry, 117997 Moscow, Russia; 2Belozersky Institute of Physico-Chemical Biology, Lomonosov Moscow State University, 119234 Moscow, Russia; 3Institute of Biochemistry, Medical Faculty, Justus-Liebig-University, 35392 Giessen, Germany

**Keywords:** IRES, translation initiation, poliovirus, PCBP2, CSDE1, UNR, STRAP, GARS, ITAF

## Abstract

The phenomenon of internal initiation of translation was discovered in 1988 on poliovirus mRNA. The prototypic *cis*-acting element in the 5′ untranslated region (5′UTR) of poliovirus mRNA, which is able to direct initiation at an internal start codon without the involvement of a cap structure, has been called an IRES (Internal Ribosome Entry Site or Segment). Despite its early discovery, poliovirus and other related IRES elements of type I are poorly characterized, and it is not yet clear which host proteins (a.k.a. IRES trans-acting factors, ITAFs) are required for their full activity in vivo. Here we discuss recent and old results devoted to type I IRESes and provide evidence that Poly(rC) binding protein 2 (PCBP2), Glycyl-tRNA synthetase (GARS), and Cold Shock Domain Containing E1 (CSDE1, also known as UNR) are major regulators of type I IRES activity.

## 1. Introduction

Most cellular mRNAs are translated by the so-called m^7^G-cap-dependent scanning mechanism [1,2], which can be described in several subsequent steps. The start codon is recognized by the 43S preinitiation complex (PIC) consisting of a 40S ribosomal subunit loaded with Met-tRNAi and eukaryotic initiation factors (eIFs). The PIC first enters at the 5′ end of mRNA via recognition of the m^7^G cap, and then the 48S initiation complex scans the 5′ untranslated region (5′UTR) in 5′-to-3′ direction. Recognition of a suitable start codon (usually AUG) is followed by the dissociation of initiation factors and 60S subunit joining to form the 80S complex [3,4].

All known eukaryotic viruses rely on the cellular translation machinery to translate their mRNAs. Some viral mRNAs do not bear an m^7^G cap and hence cannot be translated via the classical “scanning” mechanism. One prominent alternative mechanism of translation initiation, first described for poliovirus (PV), is internal ribosome entry: complex RNA structural elements located in the 5′UTR are able to recruit the components of the translation machinery and direct initiation without scanning of the entire 5′UTR. Such elements are called Internal Ribosome Entry Sites, or IRESes. In the case of poliovirus, the IRES element is composed of four structural domains—from domain II to VI, excluding domain III [5,6,7,8]. It allows cap-independent translation of the viral polyprotein initiated at AUG_743_ (position corresponding to the PV Mahoney strain). In addition, translation from the universally conserved upstream AUG codon located in the stem of domain VI (known as “cryptic” AUG_586_) gives rise to the short protein UP, which enhances virus release and is important for virus propagation in intestinal organoids [9].

Currently, viral IRES elements are divided into four (or five) classes or types which are remarkably different both in RNA secondary structure organization and in molecular mechanisms which allow them to direct translation internally [10,11,12,13]. Whereas some IRES elements require a limited number of canonical initiation factors (hepatitis C virus- or HCV-like IRESes) or even no factors at all (cricket paralysis virus- or CrPV-like IRESses) to promote internal initiation, other IRESes can require not only all canonical initiation factors but also additional host proteins (IRES *trans*-acting factors, ITAFs) for their activity [11,14,15]. Clearly, it is essential to know which ITAFs are required for a particular IRES element in order to understand how it works at the molecular level. This question is especially relevant to type I IRES elements represented by poliovirus and other picornaviruses, such as rhinovirus and other related viruses, despite the fact that poliovirus IRES was the first discovered IRES element. Here we provide an overview of old and recent results devoted to ITAFs for type I IRES elements.

### 1.1. The Concept of IRES Trans-Acting Factors

Any RNA inside the cell is expected to form a complex with RNA-binding proteins, which is then called ribonucleoprotein (RNP). In the case of an IRES element, it is expected that some IRES interacting proteins can modulate its activity. This idea stems from the observation that translation of poliovirus mRNA is inefficient in the rabbit reticulocyte lysate (RRL) cell-free system but can be strongly enhanced by the addition of HeLa cell crude lysate or its fractions [16,17,18,19]. The first attempt to identify such IRES-regulating proteins was carried out by Meerovitch and coworkers in 1989. Using the gel shift assay with a fragment of the poliovirus 5′UTR, the authors detected a single interacting protein of 52 kDa [20]. Four years later, it became possible to identify this protein as La autoantigen—an abundant RNA-binding protein [21]. Consequently, many research groups, including ours, implemented similar methodologies to discover dozens of ITAFs for different IRES elements, including the type I IRESes. We will return to some particular ITAFs later.

The major challenge is how to prove if a particular protein actually modulates IRES activity. All currently available biochemical approaches have their own, sometimes serious, limitations. Perhaps the most prominent technique to address the impact of individual proteins on IRES-mediated translation was implemented in 1996, when Pestova, Hellen and Shatsky succeeded in reconstituting 48S initiation complexes from purified components on the IRES of encephalomyocarditis virus (EMCV), a picornaviral type II IRES [22]. This approach, in particular when coupled with primer extension inhibition technique (termination of extension (TOE)-printing), not only allows the detection of initiation complex formation on the mRNA of interest but also provides sequence-specific information about on which codon exactly the initiation complex is formed. Using this approach, the authors showed that the canonical initiation factors known from cap-dependent translation initiation, except eIF4E, are necessary for initiation complex formation on the EMCV IRES and that a previously identified ITAF, the Polypyrimidine-tract binding protein (PTB), enhances 48S complex formation on the correct AUG start site. This technique, which became the “gold standard” for dissecting factor requirements of translation initiation on mRNA of interest, has finally been implemented also for type I IRES elements. When testing various previously reported trans-acting factors, it was demonstrated that the addition of only one ITAF, PCBP2, was sufficient to stimulate 48S complex formation on the poliovirus IRES and that the combination of PCBP2 with any of the other ITAFs does not promote further enhancement of complex formation [23]. The story then appeared to be completed at first sight; however, we propose that this may not be the case for the reasons discussed below. 

### 1.2. ITAFs That Interact with Poliovirus mRNA under Physiological Conditions

The main pitfall of the reconstitution system from purified components described above is its reliance on prior knowledge of which factors should be included in the reconstitution mixture. Thereby, the maximally achieved activity of the system using known components is artificially defined as 100%, not knowing what the maximal activity of the RNA in an appropriate cellular context could be. Another widely used approach to detect specific RNA binding proteins is the affinity purification of RNPs from cell lysates with immobilized RNA of interest. Usually, some non-specific RNA is included as a negative control in order to filter out non-specific RNA-protein interactions. Despite its utility, there are obvious and important limitations. Firstly, excess RNA bait can lead to the purification of major RNA-binding proteins with relaxed selectivity that perhaps would not bind the RNA of interest when a limited amount of RNA is present in the cell. Secondly, it is reasonable that concentrations of various ions (e.g., K^+^, Na^+^, Mg^2+^, Ca^2+^ etc.) in the binding buffer can have important effects on RNA-protein interactions and on intra- and intermolecular RNA-RNA interactions, but it is very hard or impossible to precisely control the concentrations of these ions due to the usually unknown corresponding concentrations that come from the cell lysate itself. Thirdly, the perhaps most important limitation is the fact that negative control RNA can, in principle, bind genuine ITAFs, which may be discarded during data evaluation just because of the wrong choice of the negative control. Finally, as the sensitivity of mass spectrometry increases dramatically, it becomes very challenging to select proteins of interest for validation (in our hands, typical RNA pull-down experiments with bound proteins detected by liquid chromatography-mass spectrometry (LC/MS) yield hundreds of proteins).

For the above reasons, it was necessary to develop an approach that will be able to identify such ITAFs in an unbiased way under physiological conditions. Essentially, this approach has been developed and implemented in case of poliovirus infection. In 2021, Aviner and coworkers [24] implemented high throughput mass spectrometry to analyze the composition of polysomes of cells infected with three RNA viruses—poliovirus and two flaviviruses, Zika virus (ZIKV) and dengue virus (DENV). All three viruses induce host protein synthesis shut-off, and at late time points of infection, only viral mRNAs remain associated with polysomes. Thus, the detection of the remodeling of polysome composition through the course of infection is a very useful source of information about ITAF functionality. Comparison of poliovirus polysome composition with that of flaviviruses is also important for the detection of selective ITAFs.

According to Aviner et al. [24], during poliovirus infection, a number of RNA binding proteins selectively accumulate in poliovirus-induced polysomes over time that is not enriched in case of flavivirus infection. These include previously reported poliovirus ITAFs: PCBP2 (or its substituent PCBP1), CSDE1 (a.k.a. Upstream of N-RAS, UNR) and its protein partner Serine/Threonine Kinase Receptor Associated Protein (STRAP, a.k.a. UNR-interacting protein, UNRIP), and glycyl-tRNA synthetase (GARS). These proteins start to accumulate in polysomes at early time points after poliovirus infection and peak at time points corresponding to host protein synthesis shut-off (3 to 4 h post-infection). Notably, this is not the case for all known RNA-binding proteins previously implicated in poliovirus translation; for example, PTBP1 was not enriched on viral polysomes during infection.

Inspired by these findings, here we focus on these four ITAFs and discuss what is known about their potential function in poliovirus translation from previous studies. It should be noted that the real number of ITAFs may be even higher; for instance, a number of new potential ITAFs were proposed by Aviner for poliovirus (e.g., HNRNPR, SRSF10 and TRA2B), but this is beyond the scope of this review. 

### 1.3. PCBP2

PCBP2, along with PCBP1 and hnRNP K, corresponds to the major poly(rC)-binding proteins. The common feature of PCBPs is the presence of three RNA-binding hnRNP K homology (KH) domains which are separated by linker sequences. PCBPs are multifunctional proteins implicated at multiple levels of regulation of gene expression [25,26]. In addition, PCBPs have a moonlighting function as regulators of iron metabolism [27]. 

The link between PCBP2 and poliovirus translation was established in 1996 when PCBP2 was identified as a protein that can interact with domain IV of the poliovirus IRES [28]. HeLa cell extracts depleted by PCBP2 were unable to efficiently translate poliovirus RNA, but both translation and production of infectious progeny virus could be restored by the addition of recombinant PCBP2 [29]. Soon, two research groups presented evidence that PCBP2 (or PCBP1) not only binds to domain IV to stimulate translation but also binds to the cloverleaf structure (domain I) of the poliovirus 5′UTR, and at this location, PCBPs are able to promote binding of viral 3CD protein [30,31,32]. The implication of PCBPs in both translation and replication processes was surprising, given that both processes cannot occur simultaneously [33]. It appeared that both PCBP2 and PCBP1 undergo proteolysis by the viral proteinases 3C/3CD during the course of poliovirus infection [34]. This cleavage occurs between KHII and KHIII domains, and it was proposed that the resulting C-terminally truncated proteins no longer participate in translation but maintain viral replication. In support of this hypothesis, the authors demonstrated that supplementation of PCBP2-depleted lysates with an uncleavable form of PCBP2 could rescue translation but not replication in vitro [35]. Recently, the structure of the complex of PCBP2 and its KHIII-truncated variant with the apical part of stem-loop IV has been investigated with cryo-electron microscopy and other methods. As may be expected, the truncated form of PCBP2 has about 10-fold lower affinity for the apical part of stem-loop IV [36]. 

PCBP2 likely plays a role as an RNA chaperone in IRES-mediated translation by modulating and stabilizing the structure of the central large domain IV of the PV IRES. PCBP2 binds close to the conserved GNRA tetraloop, and in the cryo-EM structure of the PCBP2-domain IV complex, this GNRA is protruded from the area where KHI and KHII domains are located [36]. Interestingly, mutation of the GUGA tetraloop to the sequence GACG resulted in the inactivation of the IRES, but the structure of the PCBP2-RNA complex remains unaffected [36]. This points out that GNRA may be involved in either long-range RNA-RNA interactions with other domains of the IRES or in interactions with other ITAFs, initiation factors or even with the 40S ribosome subunit. 

Apart from modulation of the RNA structure, PCBP2 can also be implicated in the recruitment of other proteins to the viral mRNA. For instance, it has been demonstrated that PCBP2 can interact with poly-A binding protein (PABP) [37], and this protein-protein interaction bridges the 5′end (through the cloverleaf) and the 3′end (through the poly-A tail) of viral RNAs to promote RNA cyclization which is necessary for genome replication. It was also reported that PCBP2 interacts with the SRp20 protein implicated in splicing regulation and that SRp20 is implicated in IRES-mediated translation [38]. In support of this mechanism, SRp20 re-localizes from the nucleus to the cytoplasm of poliovirus-infected neuroblastoma cells during the course of infection [39]. 

It should be mentioned that it is controversial whether and to what extent PCBP1 and PCBP2 can substitute for each other both at the “translation control” site (domain IV) and at the “replication control” site (cloverleaf). As the focus of this review is on IRES-mediated translation, we will just mention that in the reconstitution system using purified components, both PCBP1 and PCBP2 allow efficient 48S complex formation [23].

### 1.4. UNR and UNRIP

*CSDE1*, also known as *UNR* (Upstream of N-RAS), is a ubiquitously expressed gene that codes for an RNA binding protein implicated in posttranscriptional regulation of gene expression (see [40,41] for dedicated reviews). UNR contains five copies of the nucleic acid-binding cold-shock domain (CSD1 to 5), interspersed with four pseudo-CSDs that cannot bind to RNA. 

UNR is known to regulate the stability or translation of numerous cellular mRNAs, non-coding RNAs and viral mRNAs in opposing ways, being either activator or repressor. A number of examples include *FOS* [42], *APAF1* [43], *PABPC1* [44], *PTH* [45], *CDK11A,* also known as *PITSLRE* [46], drosophila *SXL* mRNA [47,48], and also rhinovirus and poliovirus mRNAs, which will be described in detail later on [49,50]. Such a variety of effects is perhaps explained by the different locations of UNR binding sites on the respective target RNAs and different protein partners that interact with UNR. 

One prominent protein partner which may be implicated in the UNR-dependent mechanisms of posttranscriptional control is the poly-A binding protein, PABP. There are several lines of evidence that UNR-PABP interaction is important for mRNA repression at either the translation or stability level. 

One example is found for *FOS* (a.k.a. *c-Fos*) mRNA that is remarkably unstable. It was initially demonstrated that two distinct sequence elements are responsible for rapid mRNA degradation by various pathways, and one of the instability determinants, mCRD, is located in the protein-coding region [51]. Interestingly, the destabilizing effect of the mCRD has been shown to be dependent on its translation [52]. In 2000, Grosset et al. found that the mCRD destabilizing function depends on its distance to the polyA tail and that the mCRD sequence recruits five proteins, one of which is UNR [42]. In a follow-up study, it was demonstrated that out of the five mCRD interacting proteins, only UNR directly binds to mCRD and that UNR directly interacts with PABP in an RNA-independent manner [53]. Importantly, it was further shown that UNR binding does not preclude PABP from interacting with the poly-A tail, thus allowing long-range RNA interactions between mCRD and the poly-A tail mediated by the protein-protein bridge. Finally, the authors provide compelling evidence that exactly the ribosome that translates the mCRD sequence displaces mCRD-bound proteins, and this event triggers rapid mRNA deadenylation with the involvement of CCR4, yet another UNR-binding partner [53]. Thus, in this case, we encounter a kind of “burn after reading” mechanism of mRNA stability regulation that differs from classical autoregulatory negative feedback loops (like that for PABP described below). 

Another functional role of UNR/PABP has been found in a *cis*-acting element found in the *PABPC1* mRNA 5′leader that is important for translation repression. Repression of *PABPC1* mRNA translation involves the binding of PABP to the adenine-rich autoregulatory sequence (ARS) in the 5′-untranslated region of its own mRNA [54]. The ARS consists of several stretches of oligo(A)_6–8_ separated by conserved pyrimidine nucleotides. It appeared that the ARS binds not only PABP itself but also two additional RNA binding proteins, insulin-like growth factor 2 mRNA binding protein 1 (IMP-1) and UNR [44]. While both these proteins can directly bind to the ARS RNA element, PABP has a stimulatory effect on their binding to the ARS. In turn, PABP binds to the ARS less efficiently than to poly-A tails of similar length. Therefore, newly synthesized PABP will preferentially bind to the poly-A tails of cellular mRNAs until being produced in excess. Only then excess PABP completes the functional protein complex on the ARS. Direct protein-protein interaction of PABP and IMP-1 [55] suggests their cooperative binding and a stimulative effect of IMP-1 on PABP binding to the ARS. Then, the heterotrimeric complex of UNR, IMP-1 and PABP bound to the ARS element likely serves to block the initiation of translation by obstructing scanning ribosomes [44]. While the individual contribution of UNR to the functionality of this complex appears not yet clear, it is interesting to speculate that the known UNR-PABP interaction (see above) may be involved in the additional stabilization of this repressor RNP complex and recruitment of excess PABP to the ARS independent of its canonical function on the poly-A tails of mRNAs.

Also, the UNR-mediated translation repression of *msl-2* mRNA is dependent on the UNR-PABP interaction. In Drosophila, *msl-2* mRNA is translationally repressed in females by the sex-lethal (SXL) RNA binding protein, which binds to two *cis*-acting elements in the *msl-2* mRNA 5′ and 3′ UTRs [56,57]. These *cis*-acting elements repress *msl-2* translation by different mechanisms [58]: the SXL binding sites in the 3′UTR reduce 43S PIC recruitment to the 5′UTR of *msl-2* mRNA, while the SXL binding site in the 5′UTR blocks scanning of residual PICs that escaped the above recruitment inhibition. Interestingly, the interaction of SXL with the SXL binding sites in the 3′UTR appears to be dependent on interaction with UNR and PABP. It was demonstrated that 3′UTR-bound SXL promotes the binding of UNR to an adjacent nucleotide sequence, and UNR acts as a translational corepressor [47,48,59]. Duncan et al. [60] showed that the poly-A tail is required for 3′UTR-bound SXL-mediated translation repression and that drosophila UNR and PABP, as their mammalian orthologs, directly interact with each other. Furthermore, the SXL-UNR repressor complex binds to PABP but does not interfere with eIF4G recruitment to *msl-2* mRNA—instead, the ribosome recruitment is impaired [60].

Finally, UNR can bind directly to a Renilla luciferase reporter mRNA containing an unstructured 200 nucleotides 5′UTR by virtue of its cold shock domains 2 and 4, and UNR can interact with eIF4G and PABP, thereby increasing the PABP-eIF4G interaction in cells [61]. Moreover, in an in vitro translation system using a Renilla luciferase reporter mRNA, UNR stimulates translation of the reporter mRNA irrespective if the mRNA contains a cap or a poly-A tail, with UNR binding directly to the mRNA [61]. Taken together, the multi-domain UNR protein may be involved in several different specific protein-protein and protein-RNA interactions involved in translation regulation, including poly-A-independent recruitment of the cap-binding complex, which includes eIF4G. 

Regarding UNR requirements for type-I IRES translation, pioneering research has been performed in Richard Jackson’s laboratory. Historically, it was demonstrated that HeLa ribosome salt wash could be fractionated by ion exchange chromatography into two components, A-type and B-type activity, each of which, when added individually to a reticulocyte lysate system, could stimulate type-I IRES-mediated translation [62]. First, the A-type activity was assigned to PTB [63]. In the follow-up study carried out by the same research group to search protein(s) that contribute to B-type activity, UNR protein has been identified [49]. Interestingly, CSDE1 (UNR) copurified with its protein partner STRAP (a.k.a. UNRIP), but recombinant STRAP, in contrast to UNR, did not stimulate translation driven by the HRV-2 IRES [49]. Using the HRV-2 IRES as a model, two distinct UNR binding sites have been identified: one (presumably major) binding site is located in the middle part of IRES domain II, and the second binding site is located at the bulge of domain V [64]. In strong support of the important role of UNR in type-I IRES-dependent translation, it has been shown that in unr^−/−^ murine embryonic stem (ES) cells, translation driven by type-I HRV and poliovirus IRESes was selectively suppressed in comparison to wt cells, whereas the activity of type-II IRESes (EMCV and FMDV) was not affected [50]. Importantly, the translation of type-I IRESes can be restored by the expression of UNR [50]. In our opinion, these results, along with the polysome-induced accumulation of UNR in poliovirus-infected cells [24], strongly point out a functional role of UNR in type-I IRES-dependent translation. 

STRAP, also known as UNRIP (UNR-interaction protein), which is the second protein component of type-B activity fraction, is also enriched in poliovirus polysomes [24]. While the exact role of this protein and its interaction with UNR is not yet clear, one may mention the previous observation that STRAP has homology to eIF3i, the 36 kD subunit of eIF3 (24.8% residues identical plus 31.6% similar) [49], leading to the speculation that STRAP may be involved in indirect interaction with the small ribosomal subunit by interacting with other eIF3 subunits. 

### 1.5. GARS

Many tRNA synthetases have diverse moonlighting functions in all kingdoms of life [65]. Some of such functions are linked to tRNA mimicry elements exploited by some viral RNAs [66]. We previously demonstrated that one of the tRNA synthetases, GARS, selectively and efficiently binds to the apical part of domain V of the poliovirus IRES element which contains a “gly-anticodon stem-loop” mimicry element [67,68]. Notably, this element is evolutionarily conserved in perhaps all type-I IRES containing viruses [67,69]. Point mutation of the “anticodon” completely abrogates GARS binding and strongly decreases PV IRES-driven translation both in cell-free system and in transfected cells. This is true for mutating all three nucleotides of the “anticodon” as in our initial study [67] but also for a single nucleotide mutation (unpublished data). Moreover, according to Aviner et al. [24] GARS is the only one out of all tRNA synthetases that was selectively enriched in poliovirus polysomes. This result is in good agreement with our observation that GARS selectively binds to the poliovirus IRES and is required for IRES dependent translation [67]. 

Why do the type-I IRESes appear to be selectively regulated by GARS and, given the evolutionary conservation of its “gly anticodon,” not by other tRNA synthetases? One possible reason is that GARS is not part of the cytoplasmic multi-tRNA synthetase complex (MSC), which contains eight tRNA synthetases (one of which, EPRS, possesses two activities) and three additional multifunctional proteins (AIMP1, 2 and 3) [70,71]. Therefore, an individual (or “stand-alone”) tRNA synthetase may be more accessible for binding to viral RNA than a synthetase contained in a large cytoplasmic complex of 1.2 MDa. The second reason is that GARS, at least in some cells, performs additional functions and therefore is likely produced in excess of the amounts required for its mere canonical function in the aminoacylation of tRNA. Previously, it was reported that endogenous GARS resides in granules in the neurite projections of cultured neurons and in the peripheral nerve axons of normal human tissue [72]. Then, Park et al. showed that GARS circulates in serum and can be secreted from macrophages in response to Fas ligand [73]. GARS is able to induce apoptosis of certain cancer cells. Mechanistically, extracellular GARS binds to K-cadherin, which is followed by the release of phosphatase 2A (PP2A) and the suppression of ERK signaling through dephosphorylation of ERK [73]. It was reported later that basal GARS secretion could be observed in various cell types under normal conditions, and secretion can be enhanced under stress or tissue injury [74]. High concentrations of a particular ITAF in various cells may also be advantageous for viruses to support their translation. However, it might be pure speculation if the sequestration, particularly of GARS from its cellular pool by PV RNA, may reduce the role of GARS in the production of the cellular alarm molecule diadenosine tetraphosphate (Ap4A) [75] when cellular cap-dependent translation is shut down during PV infection, and more and more GARS enzyme may run idle. 

## 2. Perspectives

In the following, we summarize the information about the structural organization of type-I IRESes and map interactions of three different ITAFs along with canonical initiation factors (Figure 1). In this course, we also extrapolate information obtained with the HRV-2 IRES on the poliovirus IRES element. 

The poliovirus IRES domain II shows interaction with UNR, and since the second UNR binding site is located at the apical part of domain V, it seems likely that RNA-protein interactions can result in the spatial approximation of these two distal RNA structures in a translation-competent RNP. Moreover, the GARS binding site on domain V is located very close to that of UNR, and we may propose that GARS can stimulate UNR binding. Interestingly, also the upstream domain II of the type-I IRES may contain a second, weaker GARS binding site (unpublished observation). As a class II tRNA synthetase, GARS is a dimer, with the two tRNA anticodon stem-loop binding sites at the outside of the dimer pointing in opposing directions [76]. This spatial arrangement of the two tRNA anticodon stem-loop binding sites in the GARS protein dimer is a perfect precondition for binding two separate GARS binding sites in the poliovirus IRES RNA. As a consequence, we may assume that (likely cooperative) binding of GARS and UNR that each connect IRES domains II and V may facilitate long-range interdomain RNA-RNA interactions. This bridging function is reminiscent of another protein acting as a “bridge” on a picornavirus IRES. PTB binds to the domains II and IV of the type-II IRES of foot-and-mouth disease virus (FMDV) and thereby stimulates FMDV translation [77]. In addition, PCBP2 interacts with the apical part of the large domain IV of the poliovirus IRES [23]. Interestingly, the eIF3 binding site appears to be located close to the PCBP2 binding site [23]. Hence, it is reasonable to propose that PCBP2 may affect eIF3 recruitment on domain IV. Finally, eIF4G binds at the lower part of domain V [78,79]. Thus, the whole domain V appears to be involved in establishing protein-RNA interactions (GARS-UNR-eIF4G from top to bottom). In combination with the interaction of the GARS dimer with the apical parts of IRES domains II and V and the interaction of the multi-domain UNR with the RNA and with eIF4G, which itself binds to the base of IRES domain V, the sum of all these interactions forms a multi-component complex that finally results in efficient recruitment of eIF4G to the 3′-region of the poliovirus IRES. eIF4G, in turn, interacts with the IRES-bound eIF3, and together they recruit the ribosomal 40S subunit to the poliovirus RNA, which then can proceed to the start codon. 

PABP may even add to the activity of this multi-component complex on the poliovirus IRES. It has been reported that PV IRES-driven reporter translation in vitro is more strongly dependent on the presence of poly-A tail than the type-II EMCV IRES-driven translation [80]. Interestingly, the addition of recombinant 2A protease, which cleaves eIF4G and, by that, removes its PABP-binding site, completely abrogates the stimulatory effect of the poly-A tail on poliovirus IRES-dependent translation [81]. Given the high dependence of poliovirus IRES-driven translation on the poly-A tail, it is reasonable to propose that additional contacts between poly-A-bound PABP and the IRES RNP may be important for efficient translation. Notably, as both UNR [53] and PCBP2 [37] can interact with PABP, we can propose that these additional interactions may be important, at least at the initial stages of viral infection, before the synthesis of viral proteins (including 2A protease) has been accomplished. After the poliovirus polyprotein has been expressed, the viral proteinases 2A and 3C become active and cleave eIF4G [82] and PABP [83]. This will then destabilize the IRES RNP complex containing eIF4G as well as reduce the binding of PABP to the 3′-terminal poly-A tail, thereby allowing binding of the uridylated VPg and the 3D polymerase to the poliovirus poly-A tail to start minus strand synthesis [84]. 

To summarize, several lines of evidence lead us to conclude that at least three ITAFs are important for poliovirus IRES activity. Previous studies established that PCBP2, UNR and GARS all can efficiently bind to poliovirus IRES, and their binding sites have been mapped to specific IRES domains. Moreover, these factors are selectively recruited to polysomes of poliovirus-infected cells. The apparent discrepancy with the results obtained so far from reconstituted in vitro-systems, where only PCBP2 has been shown to be sufficient for complex formation, can be explained by the following reasons. First, to our knowledge, the combination of all four factors (PCBP2, UNR, UNRIP and GARS) has never been tested in vitro. Second, a truncated eIF4G fragment rather than the full-length eIF4G or even the heterotrimeric eIF4F complex has been used in the in vitro-system for practical reasons, mainly originating from the difficult purification of full-length eIF4G protein. Therefore, the reconstituted system, as used so far, may not recapitulate the pioneer rounds of viral translation initiation when eIF4G is not yet cleaved. Third, poly-A and PABP were not included despite the fact that PV IRES activity is strongly dependent on poly-A and PABP in vitro. 

Finally, all ITAFs were prepared as recombinant proteins, which may not be fully active, either because of improper folding or due to the lack of eukaryote-specific post-translational modifications. Without questioning the utility of the extremely useful method of reconstitution from purified components, we argue that establishing the composition of IRES translation initiation complexes in living cells (after transfection of the RNA) or at least in a cell lysate where all required components are present is needed to better understand the mechanism of IRES-mediated translation. 

## Figures and Tables

**Figure 1 ijms-23-15497-f001:**
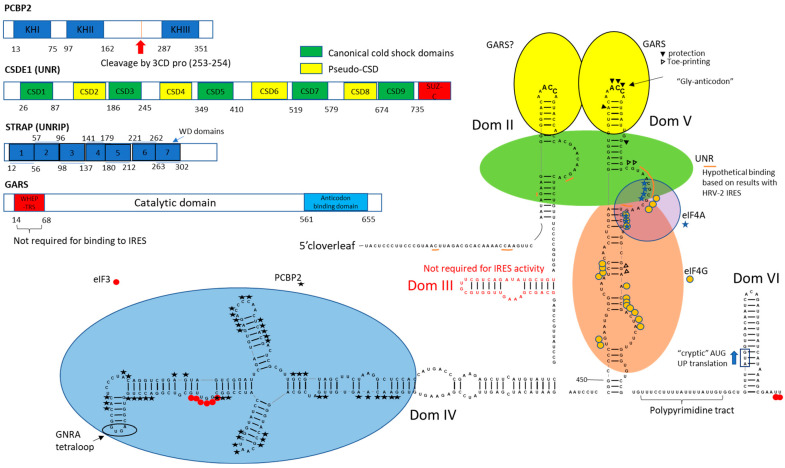
Domain organization of ITAFs involved in type I IRES-driven translation. Secondary structure of PV IRES (Mahoney strain) with highlighted functionally important sequences. Nucleotides that are protected or modified in the presence of selected proteins are marked. Note that while there is no direct evidence regarding UNR binding site on PV IRES, an approximation was made from the data obtained for the HRV-2 IRES.

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
