# Peer review of "Elusive Trans-Acting Factors Which Operate with Type I (Poliovirus-like) IRES Elements"

_ijms, 2022, doi:10.3390/ijms232415497_

Round 1

Reviewer 1 Report

A very interesting review by Andreev et al. mainly focused on the ITAFs required for poliovirus IRES translation. They have discussed some important aspects of previous research and indicated some drawbacks of previous studies. However, there are still some issues which should be addressed before it could be considered for publication.

In the first paragraph of the introduction, the authors described cellular mRNA translation. Authors are suggested to describe the formation of 40S, 43S, 48S, and 80S initiation complexes to develop the rationale of the review.

L73 Authors are suggested to write the names of the virus according to the ICTV nomenclature. “A virus name should never be italicized, even when it includes the name of a host species or genus, and should be written in lowercase. This ensures that it is distinguishable from a species name, which otherwise might be identical. The first letters of words in a virus name, including the first word, should only begin with a capital when these words are proper nouns (including host genus names but not virus genus names) or start a sentence.”

L385-86 Type I IRES also contain HRV-2 and EV-A71, which depend on many ITAFs for their IRES activity. Therefore, to claim that type I IRES depends on at least three ITAFs is incorrect. If the authors mention only poliovirus, then it is right. On the other hand, type I IRES have other viruses, as mentioned above, and many ITAFs are required for their IRES activity.

L173 “factors or even with the 40S ribosome subunit” instead of “factors or the even the 40S ribosome subunit.

Author Response

We wish to thank the editor of the special issue and both reviewers for careful revision of our manuscript. Here we provide the detailed response for reviewers’ questions.

Referee #1

A very interesting review by Andreev et al. mainly focused on the ITAFs required for poliovirus IRES translation. They have discussed some important aspects of previous research and indicated some drawbacks of previous studies. However, there are still some issues which should be addressed before it could be considered for publication.

We thank the reviewer for high evaluation of our work

In the first paragraph of the introduction, the authors described cellular mRNA translation. Authors are suggested to describe the formation of 40S, 43S, 48S, and 80S initiation complexes to develop the rationale of the review.

We modified the first paragraph of the manuscript according to reviewer’s suggestion

L73 Authors are suggested to write the names of the virus according to the ICTV nomenclature. “A virus name should never be italicized, even when it includes the name of a host species or genus, and should be written in lowercase. This ensures that it is distinguishable from a species name, which otherwise might be identical. The first letters of words in a virus name, including the first word, should only begin with a capital when these words are proper nouns (including host genus names but not virus genus names) or start a sentence.”

We thank the reviewer for spotting this mistake, it is now corrected

L385-86 Type I IRES also contain HRV-2 and EV-A71, which depend on many ITAFs for their IRES activity. Therefore, to claim that type I IRES depends on at least three ITAFs is incorrect. If the authors mention only poliovirus, then it is right. On the other hand, type I IRES have other viruses, as mentioned above, and many ITAFs are required for their IRES activity.

We agree with the reviewer that our claim that all type-I IRESes depend on PCBP2, UNR and GARS may be too strong, therefore we modified the text (lanes 393-395 of the revised manuscript) and substituted “type-I” to “poliovirus”.

L173 “factors or even with the 40S ribosome subunit” instead of “factors or the even the 40S ribosome subunit.”

Thank you, it is now corrected. 

Reviewer 2 Report

The article reviewed old and recent results devoted to ITAFs for type I IRES elements. It is suitable for publication in this magazine.

Author Response

Referee #2

I express my thanks to the editors for that requested my opinion on the manuscript entitled: „Elusive trans-acting factors which operate with type I (poliovirus-like) IRES elements“. The manuscript is a review article in which the authors describe various mechanisms for the regulation of the translational process in viruses and in particular polioviruses. The review is extensive and detailed, with the authors relying on 82 literature sources. They provided a short section where they described their own contribution to the matter, using only 2 self-citations of their own, which I think were used completely correctly and on point.

We thank the reviewer for high evaluation of our work and our practice to cite the literature in a fair way.

My remarks to the authors relate mostly to better presentation of the manuscript.

Given that this is a review article, my more significant note to the authors is that they may include more recent literary sources. Out of all 82 references, only 10 are from the last 5 years, of which 5 are from 2020-2021. As far as I can see, there are no sources from 2022. Given that it is already the end of 2022 and we are on the eve of 2023, and the topic that the manuscript examines is in an area of ​​science that has been actively worked on in recent years, there should be at least a few teams that have published their results this year. I feel that a few additions should be made to make the present manuscript appear more recent, adequate, and cited.

We believe that we were not biased towards more old manuscripts – the reason is that, unfortunately, many leading researchers who significantly contributed to investigation of poliovirus IRES driven translation are either retired or passed away. One aim of this review is to inspire a new generation of scientists to continue investigation of molecular mechanisms of IRES-mediated translation, especially because it is evident that many important questions still remain unanswered. Definitely, new technologies such as cryo-EM or high throughput mass spectrometry techniques will allow to resolve these questions in near future. Last but not the least, COVID-19 pandemics teach us that viruses, even those that are believed to be eradicated, should not be underestimated. It may be possible that next pandemic may be caused by some virus with type-I IRES, therefore investigation of poliovirus-like IRESes is of great importance. Moreover, we believe that it is possible to develop small molecular inhibitors which may act on various viruses which possess type-I IRESes.

Nevertheless, according to reviewer’s suggestion, we carefully checked the recent literature on the topic and decided to include one recent review published in 2022 from Marcelo Lopes Lastra’s group (new ref # 14).

The authors have compiled a figure in which they have presented the domain organization of ITAFs involved in the translation process driven by IRES type I, which contributes to a better understanding of the molecular mechanisms described in the manuscript.

Summarizing the knowledge thus far of the translational control mechanisms in polioviruses is essential to explain this stage of the infectious viral process. Expanding knowledge about viral protein synthesis would contribute to a better understanding of the viral replicative cycle, which could contribute to the discovery of new and more effective antiviral therapeutics.

The manuscript „Elusive trans-acting factors which operate with type I (poliovirus-like) IRES elements“ is suitable for publication in the special issue „Molecular Regulation and Mechanism of Ribonucleoprotein Complexes“.

In my opinion, the manuscript is suitable for publication in IJMS.In the manuscript submitted to me for review entitled: „Elusive trans-acting factors which operate with type I (poliovirus-like) IRES elements“, the authors review mechanisms of translational regulation in poliovirus. They present an extensive overview tracing the work of many teams on the subject, and give brief attention to their own contributions on the subject.

We, again, thank the reviewer for high evaluation of our manuscript. 

My remarks to the authors are:

  1. The keywords - although they are presented in the text what the abbreviations mean, I think it would be better to give the full names.

We are not sure if keywords should be modified in such way – it may not allow the readers to find this work as these abbreviations are usually used in search query (e.g. it is more common to search for “ITAF” rather than for “IRES trans acting factors”). In any case, we will ask the production team for advice about this issue if our manuscript will be accepted for publication.

  1. Subsections 1.3.; 1.4. and 1.5. represent only abbreviations, which does not look good. It would be nice if they were approved as subsections 1.1. and 1.2. - to be more descriptive of what is presented in them.

We are not sure that these are abbreviations – these are official protein’s names.  

  1. In the References at number 9, 34 and 71 not all authors are indicated. Let the entire author team be added to the publications.

We have used official IJMS Endnote reference style to produce reference list  - therefore it seems to be journal’s policy not to include all authors names, not our decision. If the manuscript will be accepted for publication, we will ask the production team to re-check the references style, and if it will be appropriate – to include all authors names.

  1. Only 10 of all 82 literary sources are from the last 5 years, with only 5 being from the period since 2020, and not a single literary source from 2022. This is a topic that has been increasingly worked on in recent years, so the authors can look for some even more recent sources of information, especially from the latest year 2022, which will make the manuscript even more adequate to current research on the matter.

We have added the reference to a relevant manuscript published in 2022. Our detailed response can be found above.

Reviewer 3 Report

I express my thanks to the editors for that requested my opinion on the manuscript entitled: „Elusive trans-acting factors which operate with type I (poliovirus-like) IRES elements. The manuscript is a review article in which the authors describe various mechanisms for the regulation of the translational process in viruses and in particular polioviruses. The review is extensive and detailed, with the authors relying on 82 literature sources. They provided a short section where they described their own contribution to the matter, using only 2 self-citations of their own, which I think were used completely correctly and on point.

My remarks to the authors relate mostly to better presentation of the manuscript.

Given that this is a review article, my more significant note to the authors is that they may include more recent literary sources. Out of all 82 references, only 10 are from the last 5 years, of which 5 are from 2020-2021. As far as I can see, there are no sources from 2022. Given that it is already the end of 2022 and we are on the eve of 2023, and the topic that the manuscript examines is in an area of ​​science that has been actively worked on in recent years, there should be at least a few teams that have published their results this year. I feel that a few additions should be made to make the present manuscript appear more recent, adequate, and cited.

The authors have compiled a figure in which they have presented the domain organization of ITAFs involved in the translation process driven by IRES type I, which contributes to a better understanding of the molecular mechanisms described in the manuscript.

Summarizing the knowledge thus far of the translational control mechanisms in polioviruses is essential to explain this stage of the infectious viral process. Expanding knowledge about viral protein synthesis would contribute to a better understanding of the viral replicative cycle, which could contribute to the discovery of new and more effective antiviral therapeutics.

In the manuscript submitted to me for review: "Inhibition of α1-adrenergic, non-adrenergic and neurogenic contraction of human prostate smooth muscle and stromal cell growth by the isoflavones genistein and daidzein" the studied isoflavones were isolated from legumes that are included in the daily diet intake of a large part of the population,

The manuscript Elusive trans-acting factors which operate with type I (poliovirus-like) IRES elements is suitable for publication in the special issue Molecular Regulation and Mechanism of Ribonucleoprotein Complexes.

In my opinion, the manuscript is suitable for publication in IJMS.In the manuscript submitted to me for review entitled: Elusive trans-acting factors which operate with type I (poliovirus-like) IRES elements, the authors review mechanisms of translational regulation in poliovirus. They present an extensive overview tracing the work of many teams on the subject, and give brief attention to their own contributions on the subject.

My remarks to the authors are:

1.     The keywords - although they are presented in the text what the abbreviations mean, I think it would be better to give the full names.

2.     Subsections 1.3.; 1.4. and 1.5. represent only abbreviations, which does not look good. It would be nice if they were approved as subsections 1.1. and 1.2. - to be more descriptive of what is presented in them.

3.     In the References at number 9, 34 and 71 not all authors are indicated. Let the entire author team be added to the publications.

4.     Only 10 of all 82 literary sources are from the last 5 years, with only 5 being from the period since 2020, and not a single literary source from 2022. This is a topic that has been increasingly worked on in recent years, so the authors can look for some even more recent sources of information, especially from the latest year 2022, which will make the manuscript even more adequate to current research on the matter.

Author Response

There were only two reviewers of our manuscript